# A Novel End-to-End Turkish Text-to-Speech (TTS) System via Deep Learning

Saadin Oyucu

Department of Computer Engineering, Adiyaman University, Adiyaman 02040, Turkey; saadinoyucu@adiyaman.edu.tr

**Abstract:** Text-to-Speech (TTS) systems have made strides but creating natural-sounding human voices remains challenging. Existing methods rely on noncomprehensive models with only one-layer nonlinear transformations, which are less effective for processing complex data such as speech, images, and video. To overcome this, deep learning (DL)-based solutions have been proposed for TTS but require a large amount of training data. Unfortunately, there is no available corpus for Turkish TTS, unlike English, which has ample resources. To address this, our study focused on developing a Turkish speech synthesis system using a DL approach. We obtained a large corpus from a male speaker and proposed a Tacotron 2 + HiFi-GAN structure for the TTS system. Real users rated the quality of synthesized speech as 4.49 using Mean Opinion Score (MOS). Additionally, MOS-Listening Quality Objective evaluated the speech quality objectively, obtaining a score of 4.32. The speech waveform inference time was determined by a real-time factor, with 1 s of speech data synthesized in 0.92 s. To the best of our knowledge, these findings represent the first documented deep learning and HiFi-GAN-based TTS system for Turkish TTS.

**Keywords:** speech synthesis; text-to-speech (TTS); deep learning; Turkish TTS; Turkish corpus

## 1. Introduction

The rapid advancement of technology and the proliferation of portable electronic devices has made human–machine interaction even more critical. In order to use speech information in human–machine interaction, two different core technologies need to be developed. The first is the Speech-to-Text (STT) system, which converts speech information into readable text that machines can understand, and the other is the speech synthesis (TTS: Text-to-Speech) that can address readable text. TTS systems allow audio transmission of text in digital media to users [1]. The most important application areas of the TTS systems can be listed as audiobooks, digital museums, and voice assistance systems developed to facilitate the lives of visually impaired individuals.

TTS systems are developed by combining many different disciplines, such as acoustics, linguistics, signal processing, and statistics. The main goal in TTS systems is to produce synthetic speech that is close to the naturalness of the human voice and intelligible. Intelligibility defines the clarity of the synthesized speech, while naturalness describes ease of listening. The first approaches to developing TTS systems have focused on the intelligibility of synthesized speech. However, with the development of signal processing technologies, the research objective of voice synthesis has evolved from intelligibility to naturalness. Early studies in the field of TTS showed that speech could be synthesized artificially, although it had poor intelligibility [2].

In 1791, Hungarian scientist Wolfgang von Kempelen showed that not only letters but full words could be artificially producible [3]. Kempelen developed an acoustic speech machine using a series of precision bellows, springs, bagpipes, and resonance boxes. Scientists have studied the machine developed by Kempelen until 1930. In the 1930s, the audio encoder was developed, which could automatically analyze speech based on its basic tones

and vibrations in Bell Labs. Homer Dudley developed the first electronic voice synthesizer machine, called Voder, which performed voice synthesizing on this encoder system [4]. The Voder electronic speech synthesizer is known as the first electronic machine to synthesize speech without human intervention. After switching from mechanical machines to electronic systems, Umeda and her colleagues introduced the first system to read general English text in 1968 [5].

By the early 1980s, numerous speech synthesis systems emerged for commercial use. Successful systems such as DECtalk, Whistler, and MBROLA catered to different languages. DECtalk uses a serial/parallel formant synthesizer to simulate the human audio path [6]. However, the quality of the synthesized speech is not able to meet the practical demand, as the extraction of the formant parameters is a challenging task. Therefore, the Pitch Synchronous Overlap and Add (PSOLA) algorithm was introduced to improve the quality and nature of the synthesized speech [7]. PSOLA works by dividing the speech waveform into small overlapping sections. Segments are sampled according to the fundamental frequency or period of the signal, and then the height or length of the signal is changed using the PSOLA method. The resulting segments are combined using the overlap insertion technique. Although PSOLA has been successful in regulating the prosody of the synthesized speech, the results are still not as natural as human speech. PSOLA requires the pitch period or starting point to be annotated accurately. Any error at pitch period or starting point will directly affect the quality of the synthesized speech. To overcome this problem, scientists developed Statistical Parametric Speech Synthesis (SPSS) systems [8]. The quality and naturalness of the synthesized speech with SPSS-based studies have been greatly improved. There are two methods commonly used in SPSS-based approaches. The first is Hidden Markov Model (HMM)-based [9], and the other is based on a multi-layered artificial neural network [2]. In the synthesis phase, the trained models attempt to estimate acoustic characteristics parameters. The predicted acoustic feature parameters are converted to speech using an audio encoder.

It has been observed in the literature that speech synthesis technologies have evolved over time and the quality of synthesized speech has increased. Many studies have been conducted, ranging from Formant-based parametric speech synthesis [10] to waveform unification-based methods [11]. The comprehensibility and naturalness of the synthesized speech have been greatly improved with the SPSS-based solutions that are frequently used at present. However, speech still cannot be synthesized in the naturalness of the human voice. The main reason for this situation is that the existing methods are based on non-comprehensive (simplified) models that contain only single-layer nonlinear transformation units. Related studies have shown that noncomprehensive models have good performance on data with less complex internal structures and weak constraints. However, in real life, when processing data with complex internal structures, such as speech, video, images, etc., noncomprehensive models tend to have lower success rate. Therefore, more powerful modeling capabilities should be used to effectively capture the hidden internal structures of the data and characterize the data. Many researchers have proposed deep learning-based solutions in the speech synthesis process [12,13]. Deep learning has become one of the most remarkable research fields due to its applicable learning abilities in almost every aspect of human life [14]. DL-based models have made significant advances in many areas, such as handwritten documents recognition [15,16], medical image analysis [17,18], remote sensing [19], semantic segmentation [20,21], and others [22,23]. DL-based models have made significant advances in many areas, such as handwriting recognition, machine translation, speech recognition, and speech synthesis. However, in order to use deep learning models in the field of speech synthesis, a large amount of speech-text data is needed. In resource rich languages such as English, it is easy to find the relevant dataset (corpus) [24]. However, there is no accessible corpus available that can be used in the Turkish TTS process. Therefore, our study was based on the development of a Turkish speech synthesis system at the deep learning scale. Specifically, we worked to create a sophisticated system that utilizes deep learning techniques to generate high-quality synthesized speech in Turkish.

The task of developing a DL-based TTS system is divided into two according to the characteristics of the target speaker corpus. In the first method, a large amount of corpus is obtained from only one speaker. The basic principle of this method is to train a speech synthesis system with a large amount of speech data obtained from the target speaker. By utilizing speaker adaptation or speaker encoding operations, the second approach involves developing a speech synthesis system with only a small number of examples from the target speaker. As a result, this approach requires fewer speech samples compared to the first approach. The basic idea of the speaker adaptation process is to train an acoustic model for the target speaker by fine-tuning a trained model with training data containing the data of many speakers [25]. The speaker encoding method involves utilizing a speaker encoder in TTS systems to extract speaker embedding, which characterizes the target speaker's voice and style. The main objective of developing TTS systems with this method is to capture the speaker characteristics by extracting text-independent speaker embeddings from the target speaker's voice. Commonly used speaker encoders in TTS systems include d-vector and x-vector [26]. In order for these methods to be adapted to Turkish speech synthesis systems, a large-sized speech-text corpus or pre-trained acoustic models are needed [26]. Currently, the lack of a Turkish corpus or a previously trained acoustic model available makes it difficult to process Turkish in the field of TTS.

Within the scope of this study, a large amount of corpus was obtained from one male speaker. The previously prepared Turkish texts were voiced by the selected speaker. Then, text and voice matches were checked by real users and made ready to develop the TTS system. A DL-based Turkish TTS system was developed with the resulting corpus. Tacotron 2, which includes an acoustic property prediction module and an audio encoder module, was used in the development of the TTS system [27]. The acoustic feature prediction module includes a recurrent sequence-to-sequence feature prediction network that predicts a sequence of mel-spectrogram frames from an input character string. The audio encoder module converts a character sequence into a hidden feature representation which the decoder consumes to predict a spectrogram. In this study, instead of the WaveNet audio encoder, which has a significant effect on the speech synthesis speed, Generative Adversarial Networks (GAN) were used in the audio encoding module, which was originally proposed for image production [28]. The generation of a speech sample using WaveNet is inherently slow [29]. In this study, speech was produced with the help of a High-Fidelity Generative Adversarial Network (HiFi-GAN), which is a convolutional type of network that uses mel-spectrograms as input to the GAN network. The quality of the produced speech was determined by real users using the Mean Opinion Score (MOS) [30]. In addition, the quality of the voice synthesized by means of the Mean Opinion Score-Listening Quality Objective (MOS-LQO) [31] was evaluated objectively. The first DL- and HiFi-GAN-based results in the literature are presented in this work for Turkish, based on our best knowledge. In addition, the present work provides useful information in terms of the difficulties encountered during the development of the Turkish TTS system.

The article is organized as follows. The introduction presents detailed information about TTS technology and highlights how the quality of synthesized speech has progressed from mere intelligibility to achieving a more natural speech, as evidenced by the existing literature. The Section 2 describes the architectural structure used to improve the Turkish TTS system. In the Section 3, the relevant environments are established, and corpus preparation is made for experiments carried out within the scope of the study. The Section 4 describes the experimental results and the limits of the study in detail. In the last part, the results obtained from the study are evaluated and recommendations are presented for future works.

## 2. System Architecture

DL-based approaches have proven to be the most successful methods for developing TTS systems, despite the variety of development methods used in this field over time. A typical TTS system consists of several blocks, with text data serving as input. In the first

block, text normalization operations may be applied to convert numbers or abbreviations into spoken forms. The second block involves a recurrent sequence-to-sequence feature prediction network that maps phoneme embeddings to mel-scale spectrograms, as shown in Figure 1. While various methods can be utilized in this block, Tacotron is a popular option that uses a sequence-to-sequence architecture to produce a set of frequency spectrograms. With a single neural network trained on the data, Tacotron models the production of linguistic and acoustic features, simplifying the TTS development process [27,32]. The spectrograms obtained in the second block are subsequently transformed into a waveform using a voice encoder and presented to the user.

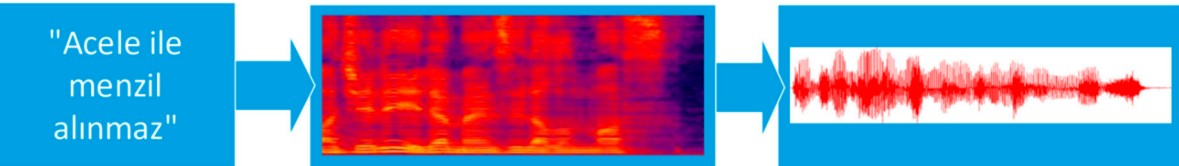

**Figure 1.** Main block diagram of TTS system.

Tacotron relies on the Griffin–Lim algorithm to convert spectrograms into speech [33], but this method produces lower sound quality than other approaches, such as WaveNet. As a result, various voice encoder approaches have been integrated into the Tacotron structure to enhance the quality of speech generated from spectrograms. Tacotron 2 is the current version of this structure, which offers a fully neural network-based approach to speech synthesis. This sequence-to-sequence model generates mel-spectrograms and utilizes a WaveNet [34] audio encoder. The Tacotron 2 architecture, shown in Figure 2, is trained directly on normalized character sequences and corresponding speech wave-forms. The resulting speech are incredibly natural-speech and often indistinguishable from human speech.

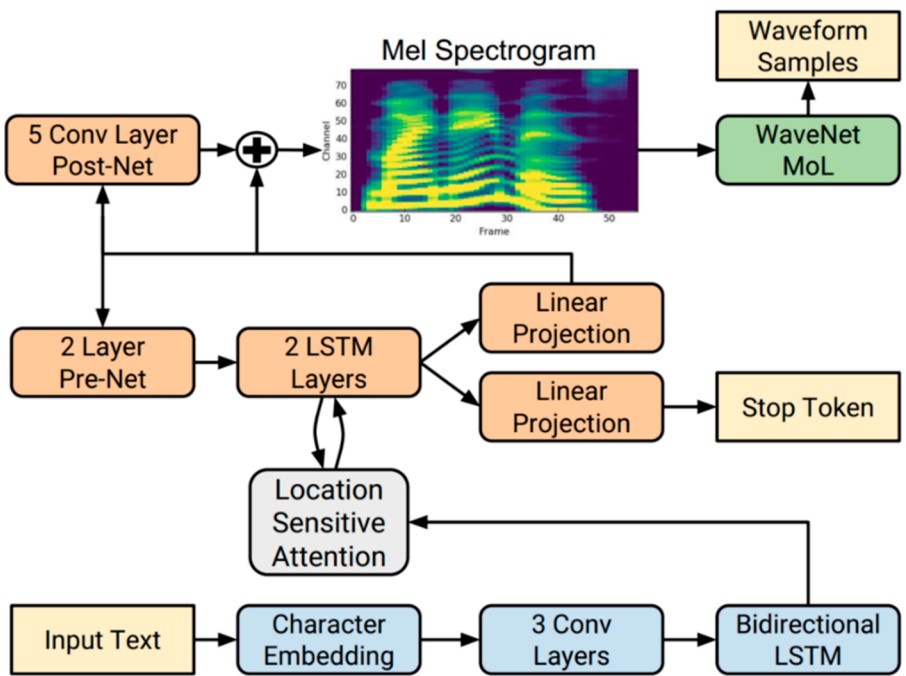

**Figure 2.** Block diagram of the Tacotron 2 system architecture [27].

As shown in Figure 2, Tacotron 2 consists of two main components. In the first component, mel-spectrograms are obtained from the input sequence and given to the feature prediction network from sequence to sequence. The second component of the TTS system includes WaveNet, which is responsible for generating time-domain waveform

samples. After the recurrent sequence-to-sequence feature prediction network generates spectrograms from the input text, these spectrograms are then transformed into time-domain waveform samples using the WaveNet.

Using a representation calculated from time-domain waveforms in the Tacotron 2 architecture requires training the two components separately. The mel-frequency spectrogram is related to the linear frequency spectrogram, namely the Short-time Fourier Transform (STFT) magnitude. It is obtained by applying a non-linear transformation to the frequency axis of the STFT, inspired by the measured responses from the human auditory system. The use of an auditory frequency scale in this way highlights details at lower frequencies that are critical for speech intelligibility. However, while linear spectrograms discard phase information, algorithms such as Griffin–Lim [33] can predict this discarded information. The Griffin–Lim structure used in Tacotron 1 makes time domain transformation possible via inverse STFT. The mel-spectrograms used in Tacotron 2 discard more information, presenting a challenging inverse transformation problem. However, when compared to the linguistic and acoustic features used in WaveNet, the mel-spectrogram is a simpler, lower-level acoustic representation of the audio signals. Therefore, it is possible to produce high-quality speech from mel-spectrograms using a WaveNet structure.

WaveNet, which is the basis of Tacotron 2's architecture, represents an auto-regressive convolutional neural network that predicts speech samples from linguistic features. Spectrograms are used instead of linguistic features as input in the WaveNet structure. The main drawback of the sound encoders prepared with the specified structure is that they can produce only one speech example at a forward pass. WaveNet is an autoregressive model that uses previous samples to generate each new sample. As a result, WaveNet needs to process the previous samples one-by-one to generate the waveform, which can increase the processing time. Many WaveNet architectures have been presented, trying to solve this problem, and streaming-based audio encoders have been proposed [35]. Models of this type predict the conditional distribution of a speech signal conditioned on acoustic characteristics. However, studies in this area have presented limited improvements. For this reason, it has been proposed to use GANs instead of WaveNet [28].

GANs are often used in TTS to enhance the realism of the speech waveforms generated by models such as WaveNet, which are commonly used for speech synthesis. The GAN architecture consists of two networks: a generator network that produces realistic-looking examples and a discriminator network that attempts to distinguish these examples from real ones. The generator network is trained until it produces speech examples of the desired quality and naturalness. Although GANs are efficient in terms of computational time, they are weaker in terms of speech quality than autoregressive models. However, to overcome this disadvantage of GANs, HiFi-GAN has been proposed [36]. HiFi-GAN can produce high-quality speech samples by processing mel-spectrograms, which represent the frequency spectrum of speech signals. However, achieving high sampling rates is necessary for producing natural and high-quality speech. Therefore, HiFi-GAN performs upsampling, increasing the speech samples to a higher sampling rate. This is achieved using a Convolutional Neural Network (CNN) structure based on the properties of the generator network. The CNN structure aims to increase the quality of the speech sample by upsampling a speech sample given at a low sampling rate with mel-spectrogram features to a higher sampling rate. Additionally, the Multi-Receiver Field Fusion (MRF) module is used in the HiFi-GAN structure to achieve high-quality sound synthesis. MRF helps to combine multiple source signals (multi-speaker) used for voice synthesis. The MRF module is used to appropriately select between source signals from multiple speakers to produce a higher-quality speech synthesis. This module also combines different characteristics of different speakers to produce a synthesized speech signal that covers a wider acoustic field. The basic manufacturer architecture is shown in Figure 3.

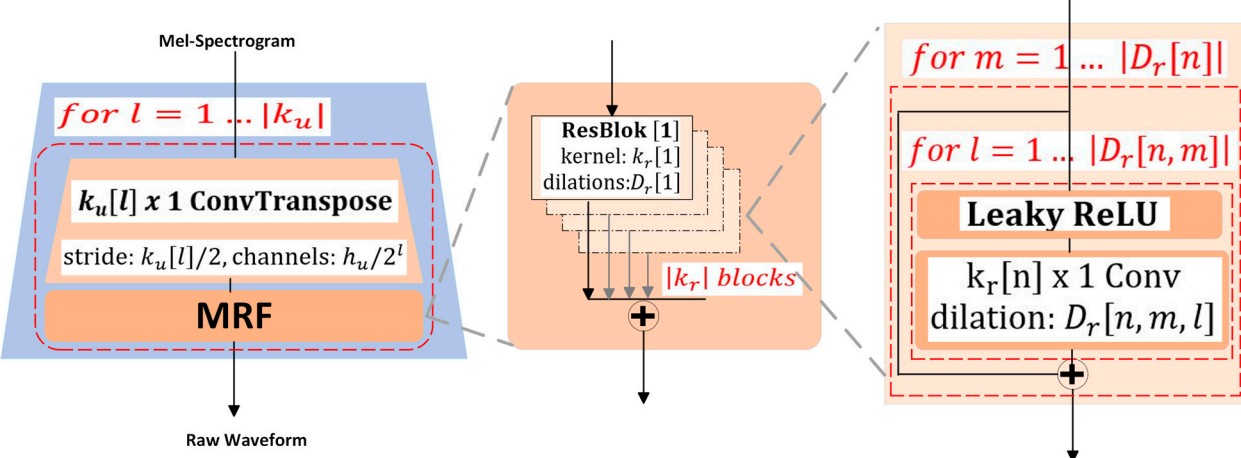

**Figure 3.** GAN generator architecture [36].

Figure 3 illustrates the architecture of the MRF module and the structure of a residual block. The MRF module has a hidden size of $|h_u|$ and uses convolutions with a kernel size of $|k_u|$ and dilation rate of $|D_r|$. To match the temporal resolution of the raw waveforms, the generator upsamples mel-spectrograms up to $|k_u|$ times. The MRF module adds features from $|k_r|$ residual blocks with different kernel sizes and dilation rates. Within the MRF module, the *n*-th residual block has a core size of $k_r[n]$ and expansion ratios of $D_r[n]$. The MRF module returns the sum of the outputs from multiple residual blocks to capture different periodic patterns underlying speech signals, which consist of sinusoidal signals with various periods. To further enhance the model's ability to capture sequential patterns and long-term dependencies, two discriminator structures are used, as proposed in [37]. The Multi-Period Discriminator (MPD) consists of several sub-separators, each processing a part of the input speech periodic signals. Meanwhile, the Multi-Scale Discriminator (MSD) evaluates speech samples at different levels sequentially. These discriminators work together with the MRF module to improve the quality of the generated speech. The proposed approach, as described in [37], can effectively identify different periodic and sequential patterns in the input data, which is essential for generating realistic-sounding speech.

The architecture of MSD is taken from the main architecture of GAN. MSD is a mixture of three sub-separators that work at different input scales. Each of the sub-separators in MSD refers to a stack of step-by-step and grouped convolution layers with Rectified Linear Unit (ReLU) activation. The discriminator size is increased by reducing the step and adding more layers. Weight normalization is applied except for the first sub-splitter, which works on raw sound. Instead, an attempt is made to stabilize training by applying spectral normalization [37]. Thus, the quality of speech synthesis is improved, and the speed of inference is reduced.

Based on Tacotron, WaveNet, and HiFi-GAN studies, whether existing TTS systems can reach human-level quality has been tested on the LJSpeech dataset. In general, studies have been conducted on FastSpeech 2 + HiFi-GAN, Glow-TTS + HiFi-GAN, and GradTTS + HiFi-GAN systems [38]. The results obtained through these systems have shown that the speech can be synthesized with high quality. More natural and understandable speech could be synthesized by implementing the WaveNet in Tacotron 2. However, the quality of speech is still not at the desired level. For this reason, HiFi-GAN audio encoder was added instead of WaveNet in the pipeline of the DL-based TTS system in the present study. In Figure 4, the pipeline of the architectural structure presented within the scope of the study is presented.

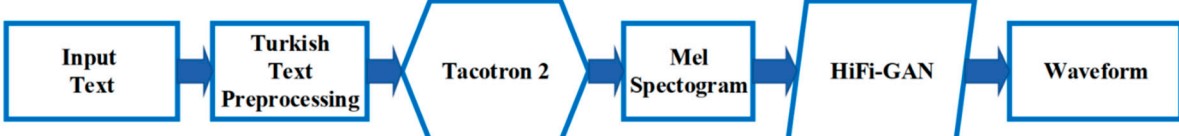

**Figure 4.** Tacotron 2 + Hi-Fi-GAN pipeline.

Tacotron 2 was used to estimate mel-spectrograms after performing pre-processing on Turkish texts. The prediction of mel-spectrograms was based on the Tacotron 2 architecture in the natural TTS synthesis in the study prepared by Shen and her colleagues [27]. Mel-spectrograms provide a visual representation of the frequency components of speech and how they change over time. However, accurately extracting mel-spectrograms can be challenging due to the nuances of speech, such as emphasis, rhythm, tone, and the presence of high or low-frequency components. Incorrectly derived mel-spectrograms can lead to issues with synthesized speech, such as incorrect intonation, distorted accents, or blending of voices. Therefore, the quality of the mel-spectrograms directly impacts the quality of the synthesized speech.

When acquiring mel-spectrograms, the number of channels is a crucial factor in obtaining a detailed visualization of the spectral content of the speech signal. Higher channel counts offer a higher frequency resolution, but this requires more processing power in the synthesis process and longer training times. Therefore, when determining the number of channels, Tacotron 2's results for different languages were considered, and a mel-spectrogram channel number of 80 was set using STFT, as in Tacotron 2 [39,40]. The STFT size was converted to the mel scale using an 80-channel mel filter bank ranging from 125 Hz to 7.6 kHz and then log dynamic range compression.

Hifi-GAN was added to the pipeline to produce a waveform from mel-spectrograms. HiFi-GAN can generate high-quality speech samples regardless of the spoken language, as it learns the common characteristics of speech signals, such as loudness, tone, and accent. This is particularly advantageous for Turkish, which has limited resources for training speech synthesis models. However, the performance of the model may vary depending on the language used for training. For example, a HiFi-GAN trained in English speech may not perform well in non-English speech. In such cases, the model needs to be fine-tuned to the target language, such as Turkish. The original Tacotron 2 architecture used WaveNet for the vocoder. However, it was suggested to use the HiFi-GAN vocoder instead of WaveNet in the present study. A GAN architecture was proposed to be used in the scope of this study. Based on the Multi-Resolution Discriminator approach proposed by You et al., each residual block was similarly combined, with two convolution layers processing patterns on the temporal axis and frequency axis, respectively [41]. In the proposed method, the first layer performs group-based convolution, which provides a very large kernel size with few parameters. The standard kernel size of the second convolution layer is 3, but an exponentially increasing expansion is present. The primary objective of this architecture is to enhance the diversity of the generator and to assess the generalizability of the multi-resolution discriminating framework. The residual net wires WaveNet-like skip connections from every residual block to the $1 \times 1$ convolution postnet [41]. Thus, a more efficient synthesis process was carried out. The Universal HiFi-GAN model was adapted for Turkish. A previously trained model for English can be used for the HiFi-GAN model in the proposed architecture. However, the waveform for Turkish may produce suboptimal results at the inference stage. Therefore, a fine-tuning of a universal HiFi-GAN model was performed with the original Turkish data. Thus, an attempt to minimize data inconsistency was made during the synthesis phase. As a result, a ready-made structure for an end-to-end Turkish TTS system is presented. Thus, the developed framework in the present study will allow the researchers to develop a new system by fine-tuning the pre-trained model.

## 3. Experimental Setup

### 3.1. Preparing a TTS Corpus

When the studies on TTS in Turkish were reviewed, it could be seen that studies are conducted to prepare syllable-based datasets and diphone-based datasets [42,43]. However, corpus-based datasets should be prepared for statistical parametric speech synthesis and DL-based speech synthesis systems. DL models are often used for processing high-dimensional data such as visual, audio, and natural language data. As the amount of data increases, these models can better understand the structural complexities in the data, leading to more accurate analyses. Therefore, incorporating large-scale corpus-based data in TTS models can significantly improve their performance. It has been observed that corpuses prepared for Turkish speech synthesis are applied to speech recognition systems and statistical parametric speech synthesis systems [44]. However, it can be seen from the literature that the corpus characteristics differ, and this difference has a direct effect on the overall success of the TTS system. Therefore, a new Turkish corpus was prepared in the present study. The features that should be considered in preparation for a corpus are listed in Table 1.

**Table 1.** Table of characteristics/units for the corpus.

| Feature | Unit |
| --- | --- |
| Language of Speech | Turkish, English, Spanish etc. |
| Total Duration of Speech | Hour |
| Category of Speech | Domain of Text |
| Speaker's Gender | Female/Male |
| Age or Age Level of the Speaker | Child, elderly, 18 and over, or Young |
| Speaker's Dialect Type | Istanbul Turkish, Aegean or Black Sea Dialect |
| Average Duration of a Speech Piece | Second |
| Minimum Duration of the Speech Piece | Second |
| Maximum Duration of Speech Piece | Second |
| Unique Word/Sentence Count | Piece |
| Total Number of Words/Sentences | Piece |

The domain of the dataset prepared in the corpus type can be everyday conversations, a literary text, newsletter content, consecutive independent texts, or a series of independent sentences that have no logical connection. These subjects should be carefully selected by diversifying. The total number of words and the number of unique words should be increased so that the corpus has a large vocabulary. Especially for Turkish, which has an agglutinative language structure, more unique words need to be voiced. Using more unique words allows the system to learn more language examples. It also makes the synthesized speech have a more natural and realistic intonation. Conversations that are paired one-to-one as text and audio recordings should be stored in short speech fragments.

While preparing the Turkish corpus, Turkish proverbs, news bulletins, reports, book chapters, religious, literary, and historical writings, names of world countries, provinces, districts, various neighborhoods and street names of the Republic of Turkey, names of various institutions and organizations of the Republic of Turkey, numbers, and a series of independent sentences with no logical connection were used. The prepared text data were voiced by a volunteer speaker who had previously received professional training. Speech recordings were obtained with the help of an Audio Technica branded studio-type cardioid condenser microphone. The most important features in choosing this microphone are its simple use, noise-free and versatile sound. A pop filter was used to reduce environmental noise and sonic booms during recording. A studio was prepared by soundproofing in the office environment for receiving recordings. The statistical characteristics of the obtained corpus are given in Table 2.

**Table 2.** Table of feature units for the Turkish corpus.

| Feature | Turkish Corpus Value |
| --- | --- |
| Total Word Count | 109.826 Pieces |
| Unique Word Count | 35.050 Pieces |
| Total Number of Characters | 745.011 Pieces |
| Total Speech Length | 12 h 38 min 59 s |
| Total Number of Speech Pieces | 8.480 Pieces |
| Average Speech Piece Length | 5.19 s |
| Minimum Speech Piece Length | 0.54 s |
| Maximum Speech Piece Length | 9.85 s |
| Average Number of Words Per Speech | 12.95 Pieces |
| Speaker's Gender | Male |
| Speaker's Age Level | Young |
| Speaker's Dialect Type | Istanbul Turkish |

The prepared corpus consists of a total of 109,826 words. Of the 109,826 words, 35,050 are unique words. This dataset, which is 12 h 38 min 59 s in total, consists of 8480 pieces of speech. The average length of speech fragments is 5.19 s, with a minimum length of 0.54 s and a maximum length of 9.85 s. The number of words per part of speech is about 13 pieces. The prepared texts were voiced by a young, volunteer male speaker. The speaker voiced the previously prepared texts by using Istanbul Turkish. The speech was recorded at 22.05 kHz mono.

*3.2. Corpus Preprocessing*

TTS systems require clear and low-noise speech samples to accurately process a specific language and accent. Long periods of silence in speech samples can reduce clarity, while noise in speech samples can disrupt the integrity of speech, making it difficult for TTS systems to process the samples correctly and resulting in lower quality synthesized speech. There should be no areas of noise and silence in the prepared corpus. Therefore, preliminary processing was carried out on the prepared corpus. VAD was used to extract quiet areas [45]. First, a binary flag was created to determine whether there was a pronunciation in the speech section. The speech signal was segmented into multiple frames and each frame was labeled based on whether it contains silence or speech. These labels were then used to group small speech samples into larger ones. The detection of the relevant gaps and noises in the speech data was performed automatically with VAD and then manually passed through real human control. In the text data, the text equivalents of the numbers were obtained. Abbreviations are not in a standard form in Turkish. Therefore, various processes were applied for different abbreviations. Abbreviations consisting of a single letter were not preprocessed. However, the pronunciation of the abbreviations read as they are written was added to the corpus. For example, the abbreviation "TRT" was added to the corpus as "te-re-te". Another type of abbreviation in Turkish is abbreviations with completely different pronunciations, which are not read as they are written. Different pronunciations of these abbreviations were added to the corpus. For example, the abbreviation "AIDS" was added to the corpus as "eydz". There was no preprocessing for abbreviations that are read as written. Finally, abbreviations in the text data were checked manually. Punctuation marks are important to be able to achieve emphasis in pronunciations. For this reason, punctuation marks were paid attention to in the texts contained in the corpus and the spaces between the letters were removed.

After the pre-processing on the corpus, the data to be used in training, testing, and model validation processes were separated. The distribution of the data is given in Table 3. The data used in the testing and verification processes were selected similarly in terms of word characteristics to make a balanced distribution. The data reserved for the test and verification process were not used in model training.

**Table 3.** Corpus distribution.

| Corpus Name | Number of Words | Number of Unique Words | Voices | Total Duration |
|---|---|---|---|---|
| Train | 107.308 | 33.230 | 8.280 | 741 min |
| Test | 1.193 | 875 | 100 | 8 min 59 s |
| Validation | 1.325 | 945 | 100 | 9 min |
| Total Data | 109.826 | 35.050 | 8.480 | 758 min 59 s |

As shown in Table 3, the lengths of the data used in the testing and verification processes are similar. The audio recording with the shortest talk time used in the testing and verification processes is 2 s. The voice recording with the longest talk time is 8 s long in the test data and 9 s long in the verification processes. The corpus distribution graph is given in Figure 5.

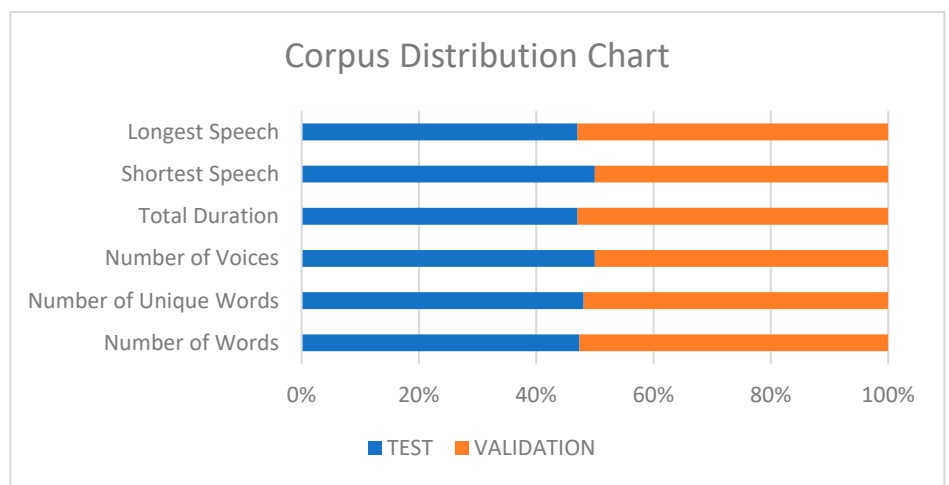

**Figure 5.** Corpus distribution chart.

### 3.3. Parameter Settings

The experiments were carried out using a single Graphics Processing Unit (GPU). The training process was performed on the Tesla P100 GPU card. Since the NVIDIA Tesla P100 contains 3584 Nvidia CUDA cores, complex models can be successfully trained. For the test, a desktop computer with an Intel i7 and 16 MB RAM capacity was used. The network architecture of the model is based on PyTorch. The parameters given in Table 4 were used to obtain mel-spectrograms from the raw audio data.

**Table 4.** Audio data parameters.

| Parameters | Value |
|---|---|
| Sampling rate | 22.05 kHz |
| Filter length | 1.024 points |
| Window size | 1.024 points |
| Number of mel-spectrogram channels | 80 |
| Mel minimum frequency | 0.0 Hz |
| Mel maximum frequency | 8000.0 Hz |

Tacotron 2 was the basis of the experimental environment prepared for the present study. However, the parameters contain corpus and Turkish-specific improvements that were prepared in our work. In summary, the prepared architectural structure used an optimized sequence-to-sequence model with a set of features that encode the sound corresponding to a string letters. Each frame calculated according to the parameters in Table 4 was converted into an 80-dimensional sound spectrogram. These spectrograms captured

not only the pronunciation of words but also the speaker's tone, loudness, and speed, as well as important points such as the emphasis on words and the intonation of a sentence according to its meaning. The encoder model parameters used for the spectrograms are given in Table 5.

**Table 5.** Model parameters.

| Parameters | Value |
|---|---|
| Initial learning rate | 0.0005 |
| Model embedding size | 512 |
| Model hidden layer size | 512 |
| Model layers | 3 |
| Batch size | 32 |
| Learning Optimizer | Adam optimizer |

The encoder model consisted of three convolutional layers, each of which contained 512 filters in the form of $5 \times 1$. These layers were followed by batch normalization and ReLU activation functions. There was an attention network that takes the output of the encoder model as input and summarizes the encoded sequence as a fixed-length context vector. The output of the convolution layer was passed through a bidirectional LSTM layer containing 512 units (256 in each direction) in order to obtain the encoded properties.

### 3.4. Training Details

The proposed system was trained on a 16 GB Tesla P100 GPU with 32 GB of RAM with a batch size of 32. The model was recorded at different epoch values to clearly see the effect of epoch values in the training process. The training process was performed from 250 to 7000 epochs. When a certain number of epoch values was reached, the relevant models were recorded and used in the test processes. The model was trained with an initial learning rate of $5 \times 10^{-4}$ and exponential decay starting at 1.5 k steps. The learning rate was reduced by 0.1 per thousand steps until it reached the minimum learning rate of $1 \times 10^{-5}$.

### 3.5. HiFi-GAN Fine-Tuning

A HiFi-GAN model trained for another language can be used to generate a Turkish speech sample from Turkish speech mel-spectrograms. However, there may be some contexts specific to Turkish, which must be learned successfully during model adjustment. For this reason, a Universal HiFi-GAN model was used, which was trained with English multi-speakers and a large-sized corpus. The used HiFi-GAN model was fine-tuned according to the audio data received from a Turkish-speaking speaker.

Aligned spectrograms of the speech samples used in the training process were created. These spectrograms helped the HiFi-GAN learn the sound of the Tacotron model. More than 5000 epochs were proposed for training HiFi-GAN. Thus, the HiFi-GAN model with 7000 epochs was fine-tuned and trained with Turkish data. In fine-tuning the HiFi-GAN model, the learning rate was set at 0.0002 and the learning decay at 0.999. The choice of learning rate in transfer learning can be influenced by the difference between the source speaker and the target speaker. Therefore, since the universal HiFi-GAN model was optimized for Turkish speakers, a small learning rate was used in the training. In addition, FFT, frequency and window size were optimized to be similar to the Tacotron 2 model.

### 4. Experimental Results

While evaluating the results of the studies conducted in the field of TTS, naturalness and intelligibility were considered. Subjective and objective evaluation methods were used in the assessment of naturalness and intelligibility. MOS or A/B testing in the literature is usually used in subjective methods. Speech quality in objective methods is generally evaluated by comparing the synthesized speech with reference speech, with no human

involvement in the comparison process, and measuring the spectro-temporal similarity of the speech signals.

A diverse dataset is essential to achieving statistically significant results with the MOS evaluation technique. Experiments should be conducted in controlled environments with specific acoustic properties and setups to ensure that each participant adheres to the same instructions or encounters similar influences. Real users were asked to rate speech recordings on a scale of 1 (bad) to 5 (excellent) to calculate the MOS value. In this study, a web interface was developed for MOS value determination, enabling users to listen to relevant conversation recordings and assign scores through the inter-face, as illustrated in Figure 6.

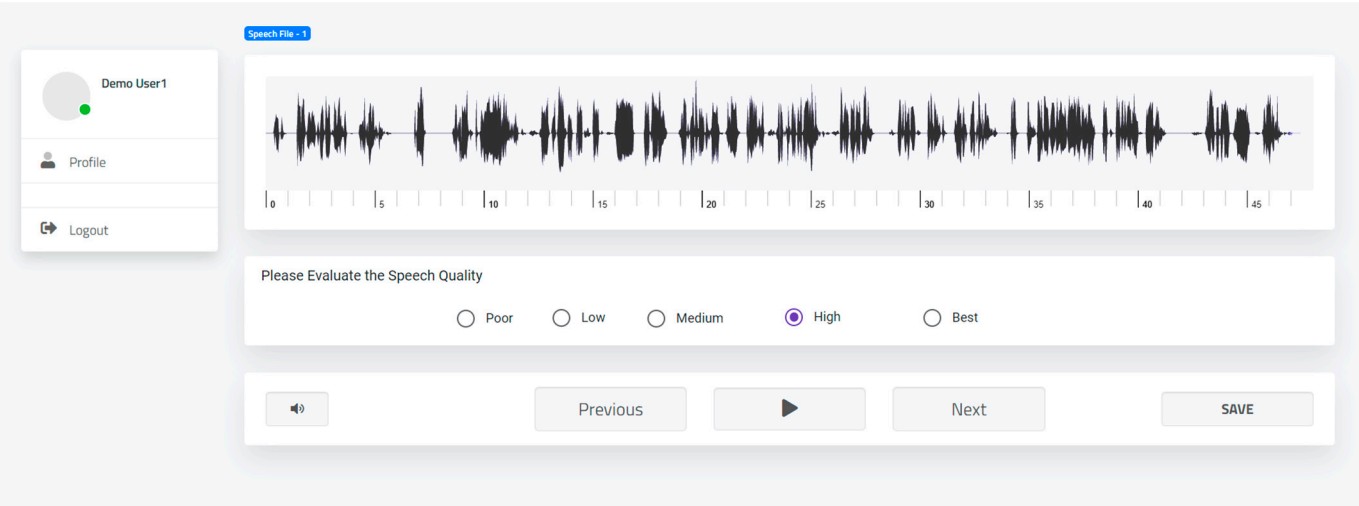

**Figure 6.** MOS evaluation interface.

Figure 6 shows the web interface where users can listen to speech recordings. Users can log in to the system with the username and password set for them and make scores. Through the developed interface, users can listen to speech recordings multiple times. After the listening process is complete, users can rate and save the recording. Twenty male and twenty female volunteer listeners were selected to perform MOS scoring via the web interface. The volunteer listeners, who were between the ages of 18 and 55, were informed before the evaluation.

Kappa statistics were utilized in this study to determine the degree of agreement among 40 different evaluators. The Kappa statistic, which is typically used for two raters, was generalized by Fleiss to measure agreement among more than two raters [46,47]. Kappa values range from −1 to +1, with positive values indicating higher agreement among raters than would be expected by chance. For each of the five different evaluation categories (Poor = 0.32, Low = 0.53, Medium = 0.68, High = 0.76, Best = 0.87), Kappa values were calculated to assess agreement among the 40 evaluators. The positive Kappa values for each category indicated a higher level of agreement among the evaluators than would be expected by chance. The lowest agreement was observed for the "Poor" category (Kappa = 0.32), while the highest agreement was observed for the "Best" category (Kappa = 0.87).

*4.1. Comparison with Human Recordings*

A comparison was made between real human speech recordings and synthesized speech recordings. Therefore, real and synthesized speech recordings were presented to the volunteer listeners by mixing. One hundred real human speech recordings and one hundred synthesized speech recordings were played to real users and they were asked to score them. The results of the MOS scoring are listed in Table 6. The synthesized speech that was played to real users was prepared with the model obtained as a result of 7000 epochs.

**Table 6.** MOS comparison between synthesized speech and human recordings.

| Human Recordings | Synthesized Speech |
| --- | --- |
| 4.61 | 4.49 |

As shown in Table 6, the synthesized speech gave quality results similar to real human speech. When the speech recordings that were given a low-quality score were examined, it was observed that although the speech intelligibility was good, there was slight noise or interference in the recording. Spectrograms were obtained to be able to observe the difference between the synthesized speech and the original speech recordings. "Acele ile menzil alınmaz" was the synthesized speech recording spectrogram of the sentence, and is shown in Figure 7, and the spectrogram of the original speech recording is given in Figure 8.

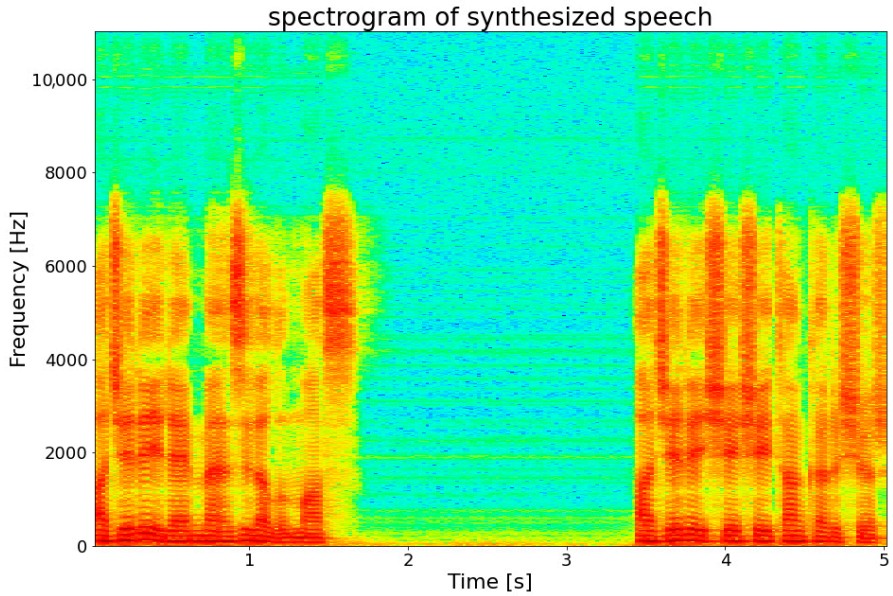

**Figure 7.** Spectrogram of the synthesized speech recording.

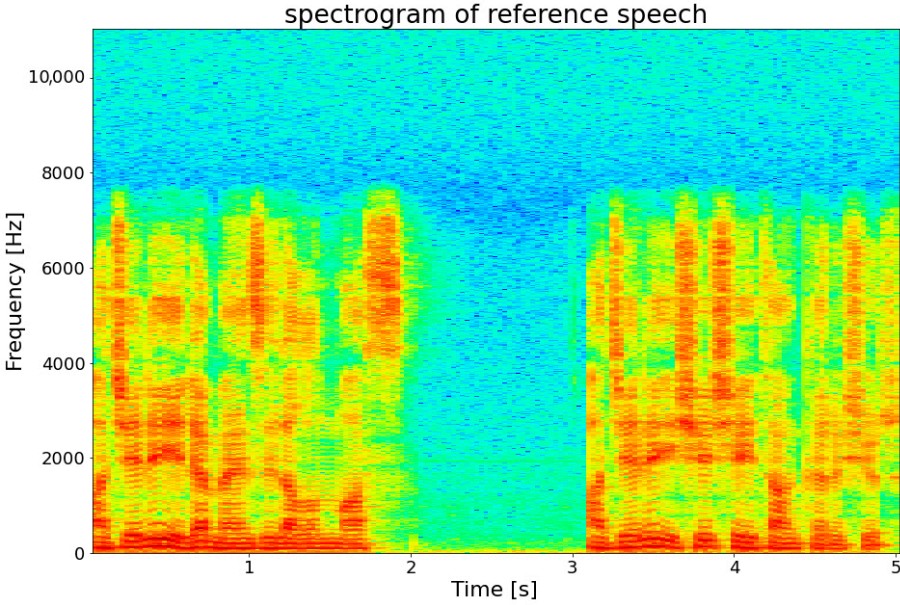

**Figure 8.** Spectrogram of the reference speech recording.

The spectrograms show how the frequency domain content of a speech signal changes over time. When the related spectrograms were examined, it was seen that similar graphs were obtained. In the same way, similarities were also seen on waveform. However, the assessments here are subjective and the perception of listening can vary from person to person. For this reason, objective MOS-LQO experiments were carried out. The VisQOL toolkit was used in the realization of these experiments [31]. VisQOL offers an objective, fully referenced measurement technique for the perceived sound quality. VisQOL, MOS-LQO score of similarity between the reference signal to produce a test file with a speech spectro-temporal uses a measure. The quality scores obtained using VisQOL are given in Table 7.

**Table 7.** MOS-LQO comparison synthesized speech.

| Epoch Level | MOS |
|---|---|
| 7000 | 4.32 |
| 6000 | 4.31 |
| 5000 | 4.29 |
| 4000 | 4.21 |
| 3000 | 4.17 |
| 2000 | 3.74 |
| 1000 | 3.70 |
| 750 | 3.53 |
| 500 | 3.51 |
| 250 | 3.44 |

It can be seen from Table 7 that the epoch value has a direct effect on MOS scoring. However, there is little improvement in the success of the model after 5000 epochs as illustrated in Table 7. It can be objectively observed from the table that there is an improvement in speech quality. The synthesized speech recordings obtained with the developed system were uploaded to Google Drive. The relevant speech files can be accessed by using the "Synthesized Speech" link in the Supplementary Materials. In addition, the Nikola Tesla documentary, voiced by the developed TTS, can be accessed via the link in the Supplementary Materials.

*4.2. Comparison with Previous Turkish TTS Systems*

The quality scores obtained within the scope of the study were compared with the previous Turkish TTS systems. For this purpose, previous Turkish TTS studies were examined in detail. The methods used in previous studies and the MOS values they obtained were investigated. The results of the Turkish TTS systems in the literature were compared with the system by using Tacotron 2 in our study. The system in our work used HiFi-GAN audio encoder in the predicted mel-spectrograms for better synthesis quality. The MOS results of the Turkish TTS studies in the literature are given in Table 8. It can be seen from the table that the results obtained in this study have better sound quality in terms of MOS than the other studies in the literature.

Corpus-based combined speech synthesis systems have been generally proposed and applied in the speech synthesis studies for Turkish. In previous studies, a Turkish phoneme set suitable and sufficient to represent all sounds in Turkish was prepared. In addition, a pronunciation dictionary was prepared for root words in Turkish in previous studies. Then, merged speech synthesis systems based on unit selection were developed. However, the developed systems have not paid attention to the prosody of Turkish. In many studies, it has been suggested to add a prosody generation module. Prosody can also be achieved with rule-based approaches. However, the rule table will be quite extensive for Turkish, which has a generative agglutinative structure. Therefore, it is not preferred.

**Table 8.** MOS comparisons between Synthesized Speech and previous TTS systems.

| References | System | Method | MOS |
|---|---|---|---|
| This Study | Deep Learning | Tacotron 2 + HiFi-GAN | 4.49 |
| [48] | Concatenation Synthesis | Corpus Based | 4.2 |
| [49] | Concatenation Synthesis | TD-SOLA | 3.85 |
| [42] | Concatenation Synthesis | Rc8660 Voice Synthesizer | 3.29 |
| [50] | Concatenation Synthesis | Statistical/Unit Selection | 3.27 |
| [51] | Concatenation Synthesis | PSOLA | 2.97 |
| [43] | Concatenation Synthesis | PSOLA | 1.78 |
| [52] | Concatenation Synthesis | PSOLA | 1.86 |

Combinational speech synthesis studies were conducted not only based on software-based, but also on hardware. The RC8660 speech synthesis platform was used in the Turkish speech synthesis system carried out by Hakan et al. [42]. The RC8660 uses the merging method, one of the TTS techniques, for the speech synthesis task [42]. The pre-recorded speech units were combined in a rule-based way. The occurrence of curtain distortions during the assembly negatively affects the success of this method. The TTS model presented within the scope of the study also took advantage of the superior success of HiFi-GAN and produced higher quality sounds. The obtained subjective and objective MOS values showed that deep learning-based methods give more successful results for Turkish.

*4.3. Speech Synthesis Latency*

The speech signals produced by the developed system have shown superior success compared to previous studies in terms of intelligibility and naturalness. However, the system as a whole is expected to voice texts quickly. Multilayer deep architectures and spectrogram transducers are inherently late responding. However, in order for the developed system to become widespread, it is expected that the synthesized speech quality will be high, and the inference time of the system will be short. Studies in the literature have focused exclusively on improving the quality of speech. Therefore, no speech synthesis latency has been found for applicable systems.

Sentences with different words were prepared to test the speech synthesis time of the developed system. A test environment with an 8-core Intel i7 process and 16 GB of RAM memory was used to measure latency times. The experimental results were evaluated using Real-Time Factor (RTF). The time taken for synthesizing a waveform of 1 s (in seconds) was obtained by RTF [53]. It was found that the RTF value was 0.92 s on average in the test conducted on 100 different speech texts. The inference time of the system lasted longer in sentences using Turkish characters. In addition, the inference time took longer in sentences using punctuation marks because the intonation changes according to punctuation marks. Google Cloud TTS, Azure TTS, which provides Turkish TTS service, was used to compare the extraction times. The comparative results of the inference periods are given in Table 9.

**Table 9.** Inference speed comparison.

| System | Latency Time |
|---|---|
| Azure TTS | 2.41 |
| Google TTS | 3.85 |
| Our TTS | 6.96 |

Turkish TTS system developed in this study and the inference time of the systems providing TTS services in Turkish are compared in Table 9. A 100-character text was used in the comparison. The waveform generated in exchange for the text is 5 s. Turkish characters and punctuation marks are included in the text. When the comparative results were considered, it could be seen that the Azure TTS system gives the best result. However, our knowledge about the background of Azure and Google TTS systems is limited. For this reason, it is unclear on which hardware the compared systems were operated. It is seen that, even in the case where the infrastructure is unclear, the system we have developed provides acceptable delay rates related to the inference speed.

*4.4. Discussion and Limitations*

Although speech synthesis presents different challenges for each language, similar technologies are being used for its development. Compared to combined speech synthesis methods, the intelligibility and naturalness of speech synthesized using SPSS-based methods are higher. However, HMM-based speech synthesis methods require the use of context decision trees to share speech parameters, limiting the naturalness of the synthesized speech. DL-based speech synthesis methods have overcome this limitation by using multiple hidden layers to map context properties to high-dimensional acoustic properties, resulting in improved speech quality. However, this process also increases the complexity of the synthesis process.

In deep TTS models, using many network parameters and hidden layers can improve synthesized speech quality, but it can also result in longer speech synthesis time. An insufficient corpus leads to overfitting, and a large amount of corpus is required for training, which can be expensive and time-consuming. High computational capability is necessary for processing a large corpus, and parallelization is used to improve network efficiency, although writing GPU code can be a time-consuming and laborious task. Intelligent programming tools are needed to facilitate TTS development in different languages. While many TTS development interfaces are available for languages with accessible corpus, there is currently no framework and corpus available for Turkish.

In languages with an additive structure, such as Turkish, the additive synthesis method has mostly been used in the speech synthesis task. According to the results obtained, it has been determined that the speech synthesis system from standard text has good intelligibility in general, but the emphasis and intonations are missing. The speech synthesized with the architecture presented within the scope of the study gave the most successful results in the available literature for Turkish. Code parts were prepared for Turkish to facilitate the TTS development process. In particular, studies were conducted to undertake the task of text normalization. The same spellings, but different meanings and readings of different abbreviations in Turkish, were determined and added to the system. The code parts that obtained the equivalent of the numbers in Turkish writing were prepared. It is expected that Turkish TTS system development studies will become widespread by sharing these frameworks.

## 5. Conclusions and Future Work

This study utilized the HiFi-GAN audio encoder in the development of a Turkish TTS system, leveraging the strengths of the Tacotron architecture. The HiFi-GAN encoder was fine-tuned for Turkish using a pre-trained universal model, resulting in synthesized speech with quality close to that of natural speech in listening tests. The achieved MOS value of 4.49 exceeded results reported in previous Turkish TTS studies. Additionally, MOS-LQO calculations, a more objective approach, produced a value of 4.32, indicating an acceptable quality for the synthesized speech. While the study focused on Turkish, the Tacotron 2 + HiFi-GAN architecture presented here can be applied to other languages.

In future studies, there is the potential to explore the synthesis of natural speech in more challenging scenarios, such as long contextual prose in audiobooks or producing sounds for songs. Researchers can benefit from the presented dataset and pre-trained model.

Moreover, future TTS studies should integrate natural language processing techniques to enhance emphasis and intonation in synthesized speech. Synthesizing the correct pronunciation of words based on their contextual meaning can improve vocalization, thereby achieving a speech quality closest to human speech.

**Supplementary Materials:** The following supporting information can be downloaded at: https://youtu.be/-Jn6z43zprQ (accessed on 21 February 2023), Video S1: Nikola Tesla documentary, voiced with the developed TTS. In this article, the Turkish TTS development codes obtained from our study can be accessed via the following link: https://github.com/saadinoyucu/Turkish-TTS (accessed on 21 February 2023). You can access the examples of "Synthesized Speech" obtained in this study from the following link. https://drive.google.com/drive/folders/187kcirx-gwQLg4nfzovhX2FNrMTz9njc?usp=sharing (accessed on 21 February 2023).

**Funding:** This research was funded by The Scientific and Technological Research Council of Turkey, grant number 121E479.

**Data Availability Statement:** You can access the Turkish corpus data obtained in this study from the following link. https://drive.google.com/file/d/1YMwTLczUs9bs-0Ukg3zlxoMrnteveiiM/view?usp=sharingAc (accessed on 21 February 2023).

**Acknowledgments:** I would like to express my gratitude to Hüseyin Polat for supporting the data collection in this study and to Münür Sacit Herdem for his valuable contributions to the development of the research scope.

**Conflicts of Interest:** The funder had no role in the design of the study; in the collection, analysis, or interpretation of data; in the writing of the manuscript; or in the decision to publish the results.

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
