# Peer review of "A Novel End-to-End Turkish Text-to-Speech (TTS) System via Deep Learning"

_electronics, doi:10.3390/electronics12081900_

Round 1

Reviewer 1 Report

The manuscript proposes a TTS system for the Turkish language using the Tacotron2 neural network architecture for speech synthesis and the HiFi-GAN, which is a convolutional type of network and uses Mel-spectrograms as input to the GAN network. The authors used the Mean Opinion Score to measure the quality of the produced speech based on the opinion of real users.

The article title is appropriate and accurately reflects the article's content. The abstract is short, clear, and well-defined. It states the main goal of the paper. The used keywords are appropriate.

The introduction is clear and correctly written. It contains a history of speech recognition theory and practice. The main ideas and applications of ML are also summarized. A corpus composed of the speech of only one male speaker has been used for training the proposed TTS system. The main goal of the study is indicated. It is also referenced with up-to-date literature sources from a suitable range of citations and covers existing relevant works. Section 2 describes the architectural structure used to improve the Turkish TTS system. In the third section, the relevant environments are established, and corpus preparation is made for experiments carried out within the scope of the study.  The experimental results and the limits of the study in detail are described in Section 4. In the last section, the conclusions of the study and future work are clear and well-written.

The manuscript content is structured correctly and contains all the relevant sections marked with subheadings. In general, the paper follows the Electronics journal’s template. The manuscript consists of 16 pages, 39 references, and 5 sections. There are 7 figures and 9 tables. All of them are well formatted and presented accurately the experimental results and proposed TTS system.  Almost 51% of all cited publications have been published in the last 5 years. The cited literature is from authoritative sources and does not need correction.

In general, the paper is formatted well and followed the Electronics journal’s template. I think that this article is suitable for the journal.

Specific comments and suggestions:

·         The terms used should be uniform. In some places in the manuscript, you write "Text-to-speech" and in others - "Text to Speech".

·         Reference 30 and 37 are the same. This is the same article presented once at a conference and a second time published in a journal. I propose removing reference 37.

Author Response

Dear Reviewer,

The authors would like to express their gratitude to the reviewer for their invaluable comments and suggestions. We have carefully considered all your valuable and constructive comments that have improved the quality and presentation of our paper. In the attached file, we would like to provide our explanations in response to the reviewer's comments. 

Kind regards

Reviewer 2 Report

This paper examines Text to speech (TTS) via deep learning (DL) for Turkish.  It proposes a Tacotron 2 + HiFi-GAN structure.

The paper is well written as far as grammar goes, i.e., uses proper English, but presents ideas poorly.  The paper contains many useful facts, but they are poorly organized.  Technical terms are noted in many places with little or no explanation.  It is highly insufficient to simply name terms, without explanation.

Specific points:

..Early studies in the field of TTS showed that a speech could be…->

 ..Early studies in the field of TTS showed that speech could be…

.. although it had poor intelligibility.. - give a citation for this claim, which is too general

..Umeda and its friends .. ->

..Umeda and her colleagues ..

..In early 1980, many .. ->

..By the early 1980s, many ..

..removing performance parameters is a challenging task.  - I do not understand this

..has been introduced .. ->

..was introduced ..

(The paper often uses the wrong tense; e.g., here, one is referring to an action well in the past, and not a continuing action)

..Segments are moved closer or further .. - a very poor way to describe this pitch period duration change

..Any error at these points.. - which “points”? 

..curtain period of starting point.  - this is surely not correct

..To overcome this problem, .. - poor idea

..natural nature .. - very poor expression

..SPSS-based approaches.  - the paper includes DL as part of SPSS, but DL does not use a statistical method

..under the guidance of linguistic characteristics.  - poorly expressed; DL for TTS does not use this

..speech synthetizations [10] .. ->

..speech synthesis [10] ..

In the second approach, it needs fewer speech samples .. - poor text presentation; describe the method first, before noting its other advantage(s)

The text around line 106 its confusing, as various technical terms are used without explanation: speaker adaptation process, fine-tuning, speaker coding method, the embedding vector, independent speaker encoder,…

..lack of a Turkish corpus .., a large amount of corpus was obtained .. - this is confusing; if there is/was a lack of corpus, how was it simply “obtained”?

..acoustic property prediction module and an audio encoder module, .. - these “modules” are not explained

..WaveNet audio encoder, which has a significant effect on the speech synthesis speed, .. - this is poorly explained; you mean that waveNet is slow?

..convolutional type of network that uses mel-spectrograms as input to the GAN network .. - none of these technical terms is explained

..goal is evolved from intelligibility to naturalness based on the literature.  - this is very confusing

..text-to-speech (TTS) .. - why redefine the acronym that has already been used many times?

..spectrograms corresponding to a set of phoneme information .. -what does this mean?  Why keep using technical terms that are not explained?

..array-to-array architecture to produce a set of frequency spectrograms. - 1) what are these “arrays”?  Is a “frequency spectrogram” different from a regular spectrogram?

Figure 1 is so general, it is useless

..WaveNet is 167 positioned, .. - what does this mean?

..conversations .. - the text repeatedly notes these, but TTS generates speech, not conversations 

..inference time of the synthesized speech is long.  - what is “inference time”?  Be specific: how long is long?

The text often refers to GANs, but never explains what they are, nor how they function.

..initially used to produce images, and later they were used for the speech synthesis task. - of what relevance is this?

The section around line 200 exemplifies the style of writing, where numerous terms are used without explanation: autoregressive models, Mel-spectrogram, HiFi-GAN structure, Up sampling, transposed convolutions, Multi-Receiver Field Fusion, residual block, sub-separators, Multi-Scale Discriminator, Rectified Linear Unit (ReLU) activation, discriminator size, Weight normalization, sub-splitter, spectral normalization.

..Mel-spectrograms directly affect the quality of the speech to be synthesized. - every step of the process affects quality.  Be more specific here.

..Therefore, as with Tacotron 2, the number of mel spectrogram channels was set to 80, .. - why? This is not clear at all.

..HiFi-GAN supports language-independent studies because it performs operations only on mel spectrograms.  - why? This is not clear at all.

..it has been suggested to use HiFi-GAN vocoder .. - who did this? Why?

..convolution structure was added between the two layers that mix the information between the channels. Thus, a more efficient synthesis process .. - why would this be more efficient?

..difon-based datasets .. -> .. diphone-based datasets 

..corpus-shaped datasets should be prepared .. -what are these?

Table 1 is poor; of what use is “hour” and “second”?  Average, max and min are all the same?

..unique number of words by diversifying .. -why would such be unique?

..Turkish, which has a productive language structure,  - what is this?

..more unique words need .. - more than what?

..total length of the conversation is one of the most important characteristics for the dataset. - yet another seemingly random statement in the text, not linked to any related ideas, or explained further.  The text throughout the paper has the semi-random style of lots of unrelated facts.

..should be no areas of noise and silence .. - why?

..speech is broken down into parts, and the speech waveforms are normalized.  -why? How?

..Abbreviations are not in a standard order .. - what does this mean?

Table 4 does not show units

..subtleties of human speech, including sound, speed and intonation.  - these are not subtleties; also, what does “sound” mean here?

..In objective methods, speech quality is generally evaluated by perceptual measurement methods.  - this seems contradictory

.. It is able to listen .. - poor English

Figs. 6-7 are not clear; why use narrowband?

..systems have not pay attention .. ->

..systems have not paid attention ..

..The speeches produced .. ->

..The speech signals produced ..

ref. 10: why say “Editor’s note: This is the ninety-sixth in a series of review and tutorial papers”?

Author Response

(The authors gave the same response as above.)

Reviewer 3 Report

This paper describes a case study, namely the application of Tacotron 2 text-to-speech synthesis (TTS) in the Turkish language. The theme is interesting. However, some additional work is needed:

1. Introduction should be focused on deep learning techniques in TTS. There is no reason to start from von Kempelen machines. Cite a textbook.

2. All discussions related to Tacotron should be gathered to a single point.

3. Fig.3 should be redrawn. Many variables inside the figure do not appear in the body of the paper.  Alternatively, the figure should be better discussed.

4. Cohen kappa statistic for the raters should be mentioned.

5. The link to data breaches the anonymity of revision.

6. Please improve research reproducibility by releasing code upon paper acceptance. A formal commitment should be undertaken.

Author Response

(The authors gave the same response as above.)

Round 2

Reviewer 2 Report

The paper has improved considerably.  The writing style is excellent.  There is likely too much technical detail, but it is OK.

minor changes:

The text repeatedly uses the term “speeches”, which should  all be replaced by “speech” (in the context of this work, “speech” is always singular; “speeches” are formal presentations)

Line 475: interface

Lines 516-518:  "Acele ile menzil alınmaz" that was the synthesized speech recording spectrogram of the sentence is shown in Figure 6, .. ->

"Acele ile menzil alınmaz" was the synthesized speech recording spectrogram of the sentence, and is shown in Figure 6, ..

Author Response

(The authors gave the same response as above.)

Reviewer 3 Report

I would like to acknowledge authors' efforts to improve their manuscript. The revised manuscript can now be accepted as is. Please include the link to code  (e.g., github link) in the camera ready paper.

Author Response

(The authors gave the same response as above.)
